# Inflammatory and Cardiac Biomarkers in Relation with Post-Acute COVID-19 and Mortality: What We Know after Successive Pandemic Waves

**DOI:** 10.3390/diagnostics12061373

**Published:** 2022-06-02

**Authors:** Catalina Lionte, Victorita Sorodoc, Raluca Ecaterina Haliga, Cristina Bologa, Alexandr Ceasovschih, Ovidiu Rusalim Petris, Adorata Elena Coman, Alexandra Stoica, Oana Sirbu, Gabriela Puha, Mihai Constantin, Gabriela Dumitrescu, Victoria Gorciac, Andrei-Costin Chelariu, Andreea Nicoleta Catana, Elisabeta Jaba, Laurentiu Sorodoc

**Affiliations:** 1Internal Medicine Department, “Grigore T. Popa” University of Medicine and Pharmacy Iasi, 700115 Iasi, Romania; ralucahaliga2017@gmail.com (R.E.H.); crisbologa@yahoo.com (C.B.); alexandra.rotariu.stoica@gmail.com (A.S.); dr.oana.sirbu@gmail.com (O.S.); gabipuha@gmail.com (G.P.); mihaiconstantin89@yahoo.com (M.C.); dumitrescu_gabriela@ymail.com (G.D.); laurentiu.sorodoc@gmail.com (L.S.); 2Second Internal Medicine Clinic, “Sf. Spiridon” Emergency Clinical County Hospital, 700111 Iasi, Romania; ovidiupetris@yahoo.com (O.R.P.); ado_coman@yahoo.com (A.E.C.); victoria_gorciac@yahoo.com (V.G.); andrei_costin24@yahoo.com (A.-C.C.); 3Nursing Department, “Grigore T. Popa” University of Medicine and Pharmacy Iasi, 700115 Iasi, Romania; 4Preventive Medicine and Interdisciplinarity Department, “Grigore T. Popa” University of Medicine and Pharmacy Iasi, 700115 Iasi, Romania; 5Rheumatology Department, Clinical Recovery Hospital, 700661 Iasi, Romania; 6Hematology Department, Regional Institute of Oncology, 700483 Iasi, Romania; 7Infectious Disease Compartment, “Sf. Spiridon” Emergency Clinical County Hospital, 700111 Iasi, Romania; catana.andreea87@yahoo.com; 8Statistics Department, FEEA, “Alexandru Ioan Cuza” University, 700506 Iasi, Romania; elisjaba@gmail.com

**Keywords:** post-acute COVID-19, 30-day mortality, inflammatory biomarkers, NT-pro BNP, high sensitivity cardiac troponin, variants of concern

## Abstract

Background: Biomarkers were correlated with mortality in critically ill COVID-19 patients. No prediction tools exist for noncritically ill COVID-19 patients. We aimed to compare the independent prognostic value of inflammation and cardiac biomarkers for post-acute COVID-19 patients and the 30-day mortality rate in noncritically ill COVID-19 patients, as well as the relation with the virus variant involved. Methods: This observational cohort study was conducted at an emergency clinical hospital between 1 October 2020 and 31 December 2021. We included consecutive patients with biomarkers determined within 24 h of presentation, followed up at least 30 days postdischarge. Results: Post-acute COVID-19 was diagnosed in 20.3% of the cases and the all-cause 30-day mortality rate was 35.1% among 978 patients infected with variants of concern. Neutrophil-to-lymphocyte ratio (1.06 [95%CI, 1.01–1.11], *p* = 0.015) and NT-pro BNP were correlated with 30-daymortality, while the monocyte-to-lymphocyte ratio (2.77 [95%CI, 1.10–6.94], *p* = 0.03) and NT-pro BNP (1.68 [95%CI, 1.00–2.84], *p* = 0.05) were correlated with post-acute COVID-19. High-sensitivity to troponin was associated with 30-day mortality (1.55 [95%CI, 1.00–2.42], *p* = 0.05). A Cox proportional-hazards model confirmed that NT-pro BNP was independently associated with mortality. NT-pro BNP remained independently associated with 30-day mortality during follow-up (1.29 [95%CI, 1.07–1.56], *p* = 0.007) after adjustment for confounders. Conclusion: Inflammation and cardiac biomarkers, determined upon admission and predischarge, in a cohort of hospitalized noncritically ill COVID-19 patients throughout successive pandemic waves, showed a predictive value for post-acute COVID-19 and 30-day mortality.

## 1. Introduction

New viral strains appeared at regular and frequent intervals during the coronavirus disease 2019 (COVID-19) pandemic. There is evidence that the Delta variant was associated with a more severe disease and poorer clinical outcomes compared with the wild-type, Alpha, and Beta variants [1]. Scientific and clinical evidence is evolving on the subacute and long-term effects of COVID-19, which can affect multiple organ systems [2]. The persistence of symptoms or development of sequelae after 3 or 4 weeks from the onset of acute symptoms of COVID-19 defines post-acute COVID-19 [3,4]. Given the variable severity of COVID-19, and its unpredictable clinical course, prognostic biomarkers would be invaluable when triaging patients to identify high-risk cases upon hospital admission or discharge.

A systemic inflammatory response is observed in COVID-19. In severe cases, neutrophilia and an increased neutrophil-to-lymphocyte ratio (NLR) was described. An elevated level of interleukin-6 (IL-6) has been observed both in mild as in severe infections with SARS-CoV-2 [5]. However, when performed on admission bloodwork, elevated serum levels of C-reactive protein (CRP), a marker of systemic inflammation, neutrophils, lymphocytes, NLR, and IL-6 had only a moderate predictive value for COVID-19, similar to routine clinical scores, such as the National Early Warning Score (NEWS2) [6].

In critically ill COVID-19 patients, defined as individuals with respiratory failure, septic shock, and/or multiple organ dysfunction [7], the use of cardiac biomarkers for risk stratification proved to be useful [8], since elevated troponin is associated with greater mortality in COVID-19, and N-terminal pro-B-type natriuretic peptide (NT-pro BNP) is strongly and independently associated with in-hospital mortality and other complications in patients with and without heart failure [9,10,11].

Studies suggest that COVID-19 patients who survive the acute phase of the disease are at risk of chronic sequelae [3]. Glucocorticoids administered in patients with severe COVID-19 pneumonia complicated with acute respiratory distress syndrome might prevent symptoms and radiological changes [12].

To our knowledge, no large-scale study has directly compared the role of inflammatory and cardiac biomarkers in predicting 30-day postdischarge mortality and post-acute COVID-19 in patients hospitalized with noncritical forms of COVID-19 throughout successive pandemic waves.

The identification of several biomarkers as independent predictors for prognostic stratification in COVID-19 patients would allow, given the wide availability and the possibility to be assessed upon hospital admission, but also during follow-up, to identify high-risk patients, to tailor medical therapy, and to guide allocation of resources, especially when these are limited.

We undertook this study to directly compare the independent prognostic value of inflammatory and cardiac biomarkers in noncritically ill COVID-19 patients infected with different variants of SARS-CoV-2, hospitalized for a medical emergency, in relation with post-acute COVID-19 and mortality.

## 2. Materials and Methods

This study is reported in line with the Strengthening the Reporting of Observational Studies in Epidemiology (STROBE) guidelines for observational studies.

### 2.1. Study Setting and Participants

This was a prospective cohort study aimed to assess the prognostic role of inflammatory and cardiac biomarkers for hospitalized patients with noncritical forms of COVID-19, based on the earliest available determination of these biomarkers upon admission and before discharge, in relation with post-acute COVID-19 and 30-day mortality. Post-acute COVID-19 is a syndrome characterized by the persistence of clinical symptoms or by the development of sequelae 3 or 4 weeks from the onset of acute symptoms, since replication-competent SARS-CoV-2 has not been isolated after three weeks [3,13]. The study was conducted in a referral emergency clinical hospital with over 125,000 Emergency Department (ED) visits annually, which was designated as a second-line hospital to assist patients with medical and surgical emergencies and associated noncritical COVID-19, during the national healthcare state of alert. Consecutive patients aged over 18, admitted to the Internal Medicine Department for a medical emergency between 1st of October 2020 and 31st of December 2021 were included after the confirmation of SARS-CoV-2 infection by RNA reverse-transcriptase polymerase chain reaction (RT-PCR) assay if they had at least one inflammatory and cardiac biomarker determination 24 h after hospital admission and if they completed a scheduled follow-up of at least 30 days after the hospital discharge (follow-up was through 31 March 2022). Upon admission, the patients had either a mild illness (i.e., various signs and symptoms of COVID-19, without dyspnea or abnormal chest imaging), a moderate disease (i.e., evidence of lower respiratory disease on clinical or imaging examination, and oxygen saturation ≥ 94% on room air at sea level), or a severe illness, defined as an oxygen saturation < 94% on room air at sea level, with tachypnea > 30 breaths/minute, a ratio of arterial partial pressure of oxygen to fraction of inspired oxygen < 300 mmHg, or lung infiltrates > 50% [7]. Patients discharged but readmitted during the clinical study period were not included in this analysis. Patients who died during hospitalization or were transferred to other departments, or the intensive care unit for developing critical COVID-19, patients who were alive postdischarge but who had not completed the 30-day follow-up before the lock of the database, patients discharged against medical advice, and patients with incomplete data were excluded. This study was approved by our Institutional Review Board, and individual written informed consent was waived based on legal standards for the national healthcare state of alert.

### 2.2. Data Collection

Routine demographics, vaccination status at time of infection, vital signs, body mass index (BMI), and comorbidities were recorded. The Charlson comorbidity index (CCI) was calculated according to the scoring system established by Charlson et al. [14] The earliest admission NEWS2 reflecting the degree of physiological dysfunction [15], routine biochemistry, and hematology results, in-hospital clinical course, and treatment were extracted from the index hospital admission using a standardized electronic data form. The levels of high sensitivity CRP (hs-CRP), complete blood count (CBC), and inflammation-related indexes based on CBC counts, natriuretic peptides (NPs), and high sensitivity cardiac troponin (hs-TnI) were collected upon ED presentation and repeated afterwards upon physician request. We also recorded the duration of hospitalization in a medical ward. These results were available to the clinician and therefore could impact on prognosis (i.e., clinicians were not blinded to these results). Interleukin-6 determination was available only in 39 patients (4%) who were transferred to our department. We analyzed NT-pro BNP into four groups: <450, 450 to 900, 901 to 1800, and >1800 pg/mL, in accordance with recommended age-related NPs cutoffs for acute heart failure (HF) diagnosis [16,17], and hs-TnI levels into three groups: undetectable (<0.05 ng/L), low (up to the 99th percentile URL), or high (≥99th percentile URL), according with the guidelines [18]. The results were obtained using PATHFAST Cardiac Biomarker Analyser (LSI Medience Corporation, Tokyo, Japan), Sysmex XT-4000i—Automated Hematology Analyzer (Sysmex Corporation, Tokyo, Japan), and ARCHITECT c16000 clinical chemistry analyzer (Abbott Laboratories, Abbott Park, IL, USA).

### 2.3. Outcome Definition 

The end point was the assessment of post-acute COVID-19 and 30-day postdischarge all-cause mortality in noncritically ill COVID-19 patients infected with the Alpha, Beta, and Delta variants of concern.

### 2.4. Statistical Analysis

Categorical variables are summarized as percentages and continuous variables as the number of nonmissing observations, the median, and interquartile range (IQR), according with the distribution of the variables. The Mann–Whitney U test or Kruskal–Wallis’ test, as appropriate, were used to identify significant differences between the outcome groups defined. The Chi-square test or the Fisher exact test, as appropriate for categorical variables, were used. For the analysis as a continuous variable, biomarker levels assessed during the hospitalization were log-transformed. All variables found to be significant in the univariate analyses for the outcomes were subjected to a multivariate logistic regression analysis, adjusted for age, sex, BMI, CCI, NEWS2 score, and cardiovascular conditions: HF, acute coronary syndrome (ACS), peripheral vascular and cerebrovascular disease, venous thromboembolism (VTE), valve disease, and atrial fibrillation (AF). Risk was expressed as hazard ratios (HRs) with confidence intervals (CIs). Receiver operating characteristic (ROC) curves were generated, and area under the curve (AUC) figures were calculated alongside sensitivity and specificity for each biomarker at Youden’s index. Survival was assessed using Kaplan–Mayer analysis and the log-rank test. The association of NT-pro BNP and hs-TnI groups with mortality was studied using Cox proportional-hazard models accounting for clinically relevant covariates (age, sex, BMI, CCI, NEWS2 score, history of AF, CKD, diabetes, hypertension, and previous HF diagnosis, vital signs upon admission, and need for supplemental oxygen). Statistical analyses were performed with SPSS (version 22.0; SPSS, Inc., Chicago, IL, USA) and STATA/SE 13.0 (StataCorp, College Station, TX, USA). All tests were two-tailed, and a *p*-value < 0.05 was considered statistically significant.

## 3. Results

During the study period, 978 patients who met the inclusion criteria and reached their 30-day follow-up were included. Patients had a median age of 69 years (range 19–94) and 502 (51.3%) were male. We recorded 509 patients (52%) with the Alpha variant in the second pandemic wave, 120 patients (12.3%) with the Beta variant in the third pandemic wave, and 349 patients (35.7%) with the Delta variant in the fourth pandemic wave (Appendix A). In Europe, the definition for an episode of reinfection is based on two positive PCR tests > 90 days apart, with at least seven symptom-free days between tests [19]. We did not record patients reinfected with a different viral strain after the first episode of infection in our cohort. In Romania, the vaccination of the general population started early in 2021. However, only 11.7% of the included patients were vaccinated. Baseline characteristics of the cohort according to the main outcomes are presented in Table 1. The median follow-up was 38 (range 22–68) days. The main reason for admission were cardiovascular emergencies (n = 567, 58%) followed by metabolic emergencies (n =131, 13.4%), acute kidney injury (n = 107, 11%), and COPD or asthma exacerbation (n = 51, 5.1%). In addition, 122 patients (12.5%) were admitted with a diagnosis of acute respiratory disease (other than asthma or COPD), other acute infection, severe anemia, and acute poisoning. The majority of patients had pneumonia or bilateral infiltrates on chest CT-scan (n = 725, 74.2%), or typical chest X-ray changes (n = 177, 18.1%). We observed that patients with the Delta variant had significantly more severe lung involvement on CT-scan (316 patients, 90.55%) as opposed to patients with the Alpha (320 patients, 62.87%) and Beta variants (92 patients, 76.66%, *p* < 0.001). The remaining patients had typical signs and symptoms of SARS-COV2 infection, consistent with mild illness [7].

The therapy administered consisted of anticoagulation in 872 patients (89.2%), corticotherapy in 590 cases (60.3%), antiviral agents in 138 patients (14.1%), and immunomodulators for 110 patients (11.2%). Hydroxychloroquine was only used for 8 patients (0.8%), and symptomatic and supportive therapy was provided for 652 patients (66.7%). A number of 343 patients (35.1%) died the next 30 days following hospital discharge (Appendix A, Table 1), and 199 patients (20.3%) developed post-acute COVID-19 symptoms (Figure 1), with systemic manifestations (39.2%), cardiovascular symptoms (23.1%), and respiratory symptoms being the most frequent (15.1%).

We did not document a reinfection with another viral strain in patients with post-acute COVID-19. The viral strain significantly influenced the occurrence of post-acute COVID-19 (Alpha variant 69.8%, vs. 30.2%; Beta variant 96.6, vs.3.4%; Delta variant 67.6%, vs. 32.4%, *p* < 0.001). However, the type of symptoms recorded (Figure 1) were not significantly correlated with the viral strain.

In terms of therapy, administration of corticotherapy was significantly correlated with protection from both post-acute COVID-19 (63%, vs. 37%, *p* < 0.001) and 30-day mortality (80.8%, vs. 19.2%, *p* < 0.001), while antiviral therapy administration was correlated with absence of long COVID (79.7%, vs. 20.3%, *p* < 0.001) and of mortality (80.7%, vs. 19.3%, *p* 0.006).

### 3.1. Inflammation Biomarkers Assessment

Significant correlations with 30-day mortality were recorded for traditional biomarkers of inflammation (i.e., hs-CRP, ferritin, presepsin) and for inflammation-related indexes: NLR, monocyte-to-lymphocyte ratio (MLR), and systemic inflammation index (SII). When analyzed based on the virus variant involved, we noticed that hs-CRP, NLR, and SII were significantly correlated with 30-day mortality in all pandemic waves. However, red cell distribution width (RDW) and white blood cells count (WBC) had this association only in patients infected with the Alpha and Delta variants, while presepsin showed a significant correlation in patients infected with the Alpha and Beta variants (Table 2).

Hs-CRP and ferritin were correlated with post-acute COVID-19 in patients recorded with the Alpha variant, while for the Delta variant, more biomarkers showed a significant correlation with this outcome, such as hs-CRP, RDW, NLR, SII, and WBC. No correlation was found for patients infected with the Beta variant, a possible explanation being the low number of patients analyzed (Table 2). The hs-CRP was significantly higher in patients who developed post-acute COVID-19 compared with survivors without sequelae (median 6.14 mg/dL, vs. 4.63 mg/dL, *p* 0.003). Moreover, we recorded significant differences in the RDW (13.97 ± 2.35, vs. 14.6 ± 2.53, *p* 0.001), NLR (6.48 ± 5.37, vs. 9.18 ± 10.05, *p* < 0.001), and SII (1687 ± 1802, vs. 2347 ± 2984, *p* 0.003) values in patients who developed post-acute COVID-19 versus asymptomatic survivors at the 30-day follow-up. The correlations between the type of symptoms recorded and the values of biomarkers are reported in Appendix A. The logistic regression (Appendix A) and AUC calculation (Table 3 and Table 4) showed that most biomarkers had modest predictive value for 30-day mortality, with NLR (AUC 0.79), MLR (AUC 0.73), presepsin (AUC 0.73), and hs-CRP (AUC 0.72) having the best performance. Other biomarkers had a weak performance (SII, neutrophils, RDW, ferritin, fibrinogen, and IL-6), with AUC figures between 0.53 and 0.69, similarly to clinical indicators.

All tested biomarkers showed a modest role to predict post-acute long COVID-19, with comparative performance being recorded for ferritin, fibrinogen, and IL-6 (Table 4).

We did not observe a significant correlation between the inflammatory markers analyzed and abnormal levels of cardiac biomarkers, irrespective of the moment of determination, upon admission or predischarge.

### 3.2. Natriuretic Peptides Assessment

A total of 795 patients (81.3%) had NT-pro BNP levels above the recommended cut-off values for hospitalized or decompensated HF [20]. The median NT-pro BNP at first determination was 803 pg/mL in patients who survived upon discharge, within the cutoff levels for defining AHF based on the median age of the subjects analyzed [16], significantly lower compared with the patients who did not survive 30 days postdischarge (Table 1, Figure 2a). Although NT-pro BNP levels were significantly higher in patients recorded with the Alpha and Delta variants who died 30 days post discharge, and in the univariate analysis, NT-pro BNP shows a predictive role, and we noticed that only the Alpha variant significantly influenced the correlation between NT-pro BNP upon admission and mortality in multivariate logistic regression analysis (Appendix A). After adjustments for inflammatory and cardiac biomarkers, vital signs, and comorbidities, multivariate logistic regression showed that NT-pro BNP predicted higher odds of death 30 days postdischarge (Appendix A). Moreover, significant correlations were observed between groups of NT-pro BNP and survival rates (Figure 2b). We also noticed a good correlation between initial NT-pro BNP levels and hs-TnI upon admission and predischarge (Appendix A).

Time-to-event analysis confirmed that NT-pro BNP concentrations upon hospital admission were significantly associated with mortality both in the whole study population (Figure 3) and in the subgroup of patients who did not have HF or who developed HF decompensations (Appendix A) (*p* < 0.001 for both comparisons by the log-rank test).

The HR and 95% CIs for variables predictive for 30-day postdischarge mortality in univariate and multivariate logistic regression are shown in Appendix A. A multivariable Cox proportional-hazards model (Table 5) confirmed that NT-pro BNP was independently associated with mortality after adjusting for all potentially relevant confounders (HR 1.63 [1.13–1.44] per logarithmic unit, *p* < 0.014).

Additionally, complementary analyses to consider further adjustment for hs-TnI (Appendix A) and D-dimer (among the subgroup of patients in which this biomarker was available, Appendix A) showed that NT-pro BNP remained independently associated with all-cause mortality during follow-up (HR 1.29 [1.07–1.56] and HR 1.36 [0.99–1.89] per logarithmic unit, respectively).

When we analyzed NT-pro BNP in relation with post-acute COVID-19, we found that this biomarker predicted higher odds for post-acute COVID-19 in the logistic regression analysis (Appendix A), but with a modest performance (Table 4).

We noticed that 615 patients (62.9%) had BNP levels above the cut-off values for HF. However, BNP did not show a significant association with poor outcomes pattern (Table 1).

### 3.3. High-Sensitivity Troponin

Among the present study participants, hs-TnI was assessed in all patients upon admission. A second value was determined predischarge, after a median of 8 days (range 5–14). A total of 699 patients (71.5%) fulfilled the criteria for myocardial injury [18] during the initial assessment and 767 patients had a second hs-TnI value consistent with myocardial injury (78.4%). When we analyzed the patients based on the virus variant involved, we did observe a correlation between admission hs-TnI and 30-day mortality in patients infected with the Alpha and Delta variants (Table 2). However, ACS was recorded only in 17 patients (3.3%) out of 509 patients infected with the Alpha variant and in 6 patients (1.7%) out of 349 patients infected with the Delta variant. Moreover, the occurrence of ACS during hospitalization was not significantly correlated with the virus strain recorded. For both hs-TnI determinations, there was a correlation with increasing NT-pro BNP levels (Appendix A). We also observed a significant correlation between predischarge hs-TnI and 30-day mortality (Figure 4, Table 3).

Predischarge high sensitivity cardiac TnI was significantly increased in patients infected with the Delta variant and correlated with post-acute COVID-19 (Appendix A). However, the statistical significance of this correlation was not obtained when all patients were analyzed and hs-TnI showed a weak predictive role for post-acute COVID-19 (Figure 4, Table 4).

## 4. Discussion

COVID-19 remains a clinical challenge for every practitioner. The World Health Organization mentioned that, among SARS-CoV-2 variants of concern, are Alpha (B.1.1.7), Beta (B.1.351), Gamma (P.1), and Delta (B.1.617.2). Alpha, Beta, Gamma, and Delta variants are all more serious than the wild-type virus in terms of hospitalization and mortality, with the Beta and Delta variants having a higher risk than the Alpha and Gamma variants [21]. Patients in our cohort infected with the Beta and Delta variants were older than the patients infected with the Alpha variant. Old age was significantly associated with 30-day mortality in patients infected with the Delta variant. These observations are in accordance with other reported data [22]. Our study showed that, irrespective of the virus variant analyzed (Alpha, Beta and Delta), some inflammation biomarkers (i.e., hs-CRP) and inflammation-related indexes (i.e., NLR, SII), as well as NT-pro BNP, were predictive for 30-day mortality. However, hs-troponin was significantly predictive for 30-day mortality only in patients infected with the Alpha and Delta variants. The majority of patients who present to the hospital will recover, but some of them rapidly develop a critical disease. After discharge, it is now apparent that clinical sequelae (long COVID-19) may persist after acute COVID-19, but their nature, frequency, and etiology are poorly characterized. An association between the female gender and long COVID-19 risk, as well as the association between presence of comorbidity, increased age, and minority ethnicity, with long COVID-19 and long COVID-19 risk were reported [2]. We also observed the association of increased age with post-acute COVID-19. However, we did not find the association of female gender and comorbidities (assessed using CCI) in our cohort. We had no minorities in the population analyzed. Moreover, we documented post-acute COVID-19 in a lower percentage of patients compared with earlier studies which included a smaller number of outpatients with mild or moderate COVID-19 [23]. The most common symptoms recorded in our cohort were consistent with other recent reports [24,25]. Systemic and cardio-respiratory symptoms were prevalent in our cohort, similar with the results reported in a recent meta-analysis, which reported the persistence of at least one symptom in 16.4% patients with mild COVID-19 and in 49.5% patients with a moderate disease [26]. Since the pathophysiologic mechanisms of long post-COVID-19 symptoms are still unclear, the biomarkers that might predict this deterioration would be invaluable when triaging patients on hospital admission to inform who can be safely discharged versus those who might need intensive care support in the near future, and a close follow-up postdischarge. There is no definitive information regarding the predictive role of inflammation and cardiac biomarkers in long COVID-19 or the long-term mortality after COVID-19. This paper presents a prospectively recruited Romanian cohort of patients with noncritically ill COVID-19 with targeted biomarker sampling at presentation. The focus of this study was the additional role of blood biomarkers when initially assessing patients presenting with COVID-19, and their predictive role, when assessed predischarge, for developing long COVID-19 and mortality.

Systemic inflammation, as measured by CRP, is strongly associated with critical illness and in-hospital mortality in COVID-19 [27]. Higher neutrophils numbers and NLR with a lower lymphocyte count were observed in severe cases of COVID-19 compared to nonsevere cases [28]. NLR is correlated with CRP and D-dimer level, therefore, NLR may serve as a reliable, cost-effective, and practical inflammatory biomarker for differentiating severe and nonsevere COVID-19 inpatients [29]. Our results showed that an association of routine inflammation biomarkers, including NLR, MLR, WBC, and hs-CRP, was correlated with 30-day mortality in noncritically ill COVID-19 patients.

In association with clinical observations, the kinetic measurement of IL-6 during SARS-CoV-2 infection is a crucial tool to predict the prognosis and outcome of patients with COVID-19 [30]. While inflammatory cytokines such as IL-6 could underpin some of the long-term neuropsychiatric features of COVID-19 [31], persistently elevated levels of IL-6 and other cytokines may be a hallmark of post-acute sequelae of COVID-19 [32]. However, these biomarkers are not routinely available for every inpatient, and several other inflammatory biomarkers might offer reliable information. It is notable that in our study, easily accessible biomarkers (i.e., ferritin, fibrinogen) showed comparative performance with IL-6 and some of the blood-based biomarkers tested for post-acute COVID-19, while for 30-day mortality, outperformed Il-6 (i.e., NLR, MLR, hs-CRP). An explanation might reside in the low number of subjects with available IL-6 determinations in our cohort.

In patients hospitalized with COVID-19, mild elevations in cardiac troponins reflecting cardiomyocyte injury and/or increased NPs concentrations as a consequence of hemodynamic stress are in general the result of pre-existing cardiac disease and/or the acute stress related to COVID-19 [33,34]. In noncritically ill patients with COVID-19, elevations up to three times the ULN are in general well explained by the combination of possible prior cardiac disease and the acute cardiomyocyte injury related to COVID-19. The level of those biomarkers correlates with disease severity and mortality [34,35]. It was suggested that persons infected with the SARS-CoV-2 Delta variant may be at a higher risk for adverse outcomes compared with those infected with other variants of concern [36], although much still remains unknown. Indeed, we noticed a significantly higher 30-day mortality in patients with the Delta variant infection as opposed to the other SARS-CoV-2 variants of concern analyzed. A possible explanation might be the older age and the higher percentage of patients with severe lung involvement.

The present study results support the hypothesis that both NT-pro BNP and hs-TnI are highly associated with 30-day mortality in noncritically ill COVID-19 hospitalized patients, consistent throughout successive pandemic waves involving different virus variants. A large nationwide observational cohort study of the American Heart Association’s COVID-19 Cardiovascular Disease Registry showed that elevations in NT-pro BNP on admission to the hospital for COVID-19 predict worse clinical outcomes, including increased risk of death and major cardiovascular complications [6]. However, this study did not report data throughout the spectrum of COVID-19, or any relations regarding the virus variant involved.

After controlling for comorbidities, presenting vital signs, clinical characteristics, and therapy provided, NT-pro BNP elevations independently predicted a 63% increase in 30-day mortality. After further adjustments for other biomarkers, increase in NT-pro BNP per logarithmic unit independently predicted 29% higher odds of death, while hs-TnI per logarithmic unit predicted a 55% higher risk of 30-day mortality. Echocardiographic reports were lacking for the majority of patients included in the present analysis, since the use of noninvasive imaging modalities was restricted by international societies during the initial phase of the pandemic [37]. As a result, we grouped our patients based on the cutoffs of NT-pro BNP for diagnosing HF [17]. In doing so, our results show that this association between NT-pro BNP levels and mortality remains independent of the development of acute HF during hospital admission or the history of chronic HF. We did not find the same predictive value of BNP, as it was previously reported in a meta-analysis [38]. In line with other reported data, BNP did not show a predictive value for the outcomes in noncritically ill survivors versus non-survivors [39]. This may be explained by the fact that, compared to BNP, NT-pro BNP and hs-TnI appeared to be better performers in predicting mortality for inpatients [9,39]. NT-pro BNP acts as a more sensitive prognostic biomarker after admission [40]. The extent to which biomarker levels reflected pre-existing cardiovascular disease or COVID-19 severity, and the dependency of natriuretic peptides on the time interval from the beginning of symptoms, are still unclear for COVID-19 patients.

Since we have not seen a correlation between inflammation and cardiac biomarkers in noncritically ill COVID-19 patients with respect to the outcomes analyzed, we propose that multimarker assessment should be provided before discharge to detect high-risk patients for developing post-acute COVID-19 or death.

The size and diversity of the medical conditions in subjects from our cohort enhances the strengths of our study in characterizing the association between inflammation and cardiac biomarker elevations and poor outcomes in noncritically ill COVID-19 patients. Thus, it is reasonable to include these biomarkers in the patient’s diagnosis, triaging, treatment, and prognosis, while recognizing that their abnormality may not be related to direct cardiovascular involvement.

Given the strength of the signal in this representative data set, and the size and diversity of the medical conditions in our cohort, we believe our study demonstrates the utility and prognostic value of elevated biomarkers of inflammation and cardiac biomarkers levels in predicting 30-day mortality and post-acute COVID-19 in hospitalized noncritically ill COVID-19 patients irrespective of the virus variant involved.

As with any observational study, our study is limited by the scope and depth of data collected. A second limitation of this study is the relatively limited sample size, especially for infections with the Beta variant, which could lead to imprecise estimates of biomarker performance. A third limitation concerns the results obtained from only a single region of Romania, which should be confirmed in larger international studies. Fourthly, the analysis was performed with baseline biomarkers and with a limited set of serial measurements and number of IL-6 determination, which, had they been available for all patients, could have been extremely relevant to better understand the relationship between noncritically ill COVID-19 patients, long COVID-19, and survival.

## 5. Conclusions

Inflammation and cardiac biomarkers, when performed on admission and predischarge, in a cohort of noncritically ill COVID-19 hospitalized patients throughout successive pandemic waves, showed a moderate predictive value for 30-day mortality and modest predictive value for post-acute long COVID-19. Age and routine clinical scores (i.e., CCI and NEWS2 score) also showed a poor predictive role. Among inflammation biomarkers, NLR, hs-CRP, and MLR had the best performance. Moreover, NT-pro BNP showed a good predictive role for 30-day mortality, irrespective of the variant of concern involved. These biomarkers can be used as tools to identify high-risk non-critically ill COVID-19 hospitalized patients with additional comorbidities. Assessing them may represent a surrogate of invasive monitoring in a context of poor resource setting, may support the tailoring of medical therapy, and guide the allocation of available resources. Further large prospective studies should validate the additional value of these biomarkers compared to routinely collected clinical information.

## Figures and Tables

**Figure 1 diagnostics-12-01373-f001:**
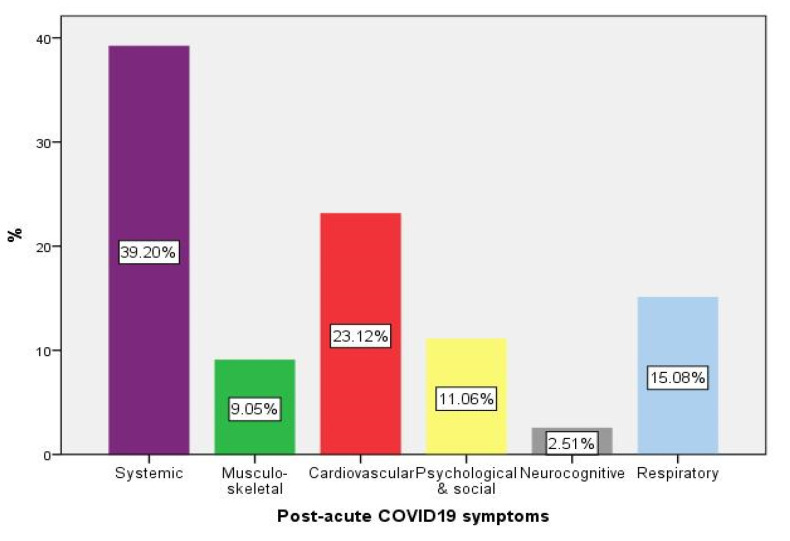
Post-acute COVID-19 in the cohort analyzed.

**Figure 2 diagnostics-12-01373-f002:**
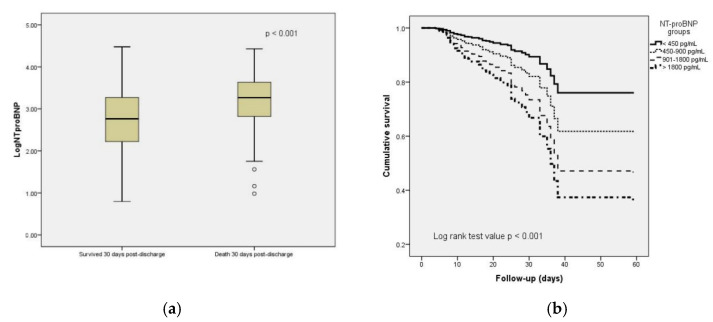
(**a**) Box plot demonstrating the logarithmic concentrations of NT-pro BNP in relation with 30-day mortality; °, represent outliers; (**b**) Cox proportional-hazards model assessing the relationship between NT-pro BNP groups and survival during follow-up.

**Figure 3 diagnostics-12-01373-f003:**
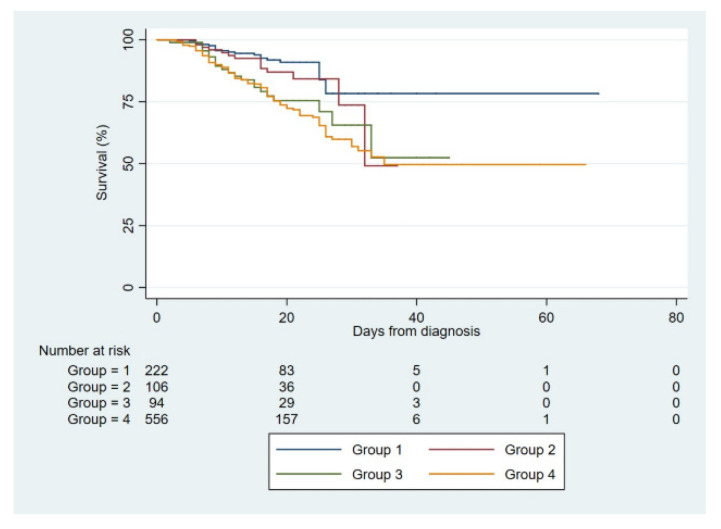
Kaplan–Meier survival curves regarding all-cause 30-day mortality according with NT-pro BNP groups in the whole study population.

**Figure 4 diagnostics-12-01373-f004:**
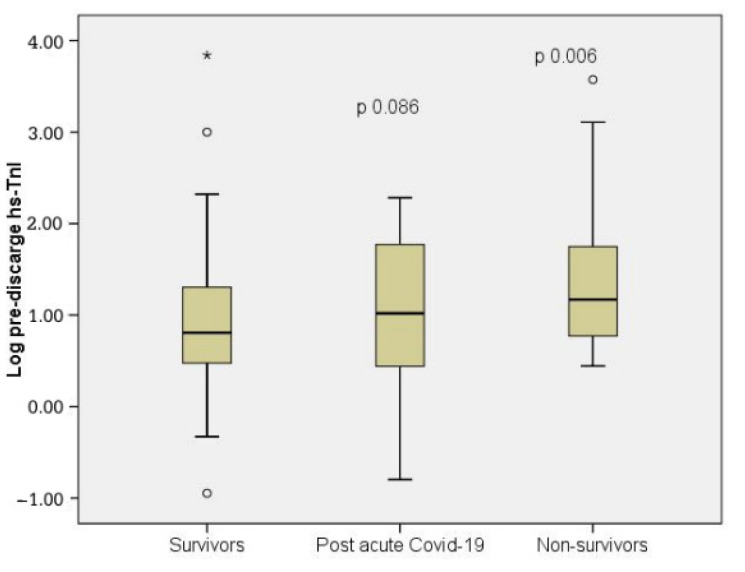
Boxplot demonstrating the logarithmic concentrations of predischarge high-sensitivity troponin I (hs-TnI) among the main outcome groups after 30 days follow up; °, represent outliers; *, represent extreme values.

**Table 1 diagnostics-12-01373-t001:** Baseline characteristics of the cohort according to the outcomes.

Variable	Total Patients N = 978	Survivors without Sequelae N = 436 (44.6%)	Post-Acute COVID-19 N = 199 (20.3%)	30-Day Mortality N = 343 (35.1%)	*p*-Value *
Virus variant, N (% **)					<0.001
Alpha	509 (52)	279 (28.5)	104 (10.6)	126 (12.9)	
Beta	120 (12.3)	49 (5.0)	29 (3.0)	42 (4.3)	
Delta	349 (35.7)	108 (11.0)	66 (6.8)	175 (17.9)	
Age, median [IQR], y	69 [59–78]	68 [56–77]	66 [55–73]	72 [65–81]	<0.001
Males, N (%)	502 (51.3)	228 (23.3)	93 (9.5)	181 (18.5)	0.703
Residence (rural), N (%)	445 (45.5)	216 (22.1)	81 (8.3)	148 (15.1)	0.002
CCI, median [IQR]	4 [2–5]	3 [2–5]	3 [2–4]	4 [3–6]	<0.001
Onset-to-admission < 7 d, N (%)	615 (62.9)	294 (30.1)	126 (12.9)	195 (19.9)	0.034
NEWS 2, median [IQR]	6 [4–8]	5 [3–6]	6 [4–7]	8 [6–10]	<0.001
SaO_2_ < 90%, N (%)	341 (35.0)	83 (8.5)	60 (6.2)	198 (20.3)	<0.001
SBP (mmHg), median [IQR]	130 [120–149]	128 [115–135]	140 [120–160]	146 [126–164]	0.114
HR (bpm), median [IQR]	85 [75–100]	86 [70–92]	82 [75–100]	93 [81–100]	<0.001
Ht (%), median [IQR]	38.9 [34.3–42.0]	39 [33.4–42.1]	39.6 [35.5–41.9]	39.8 [36.0–43.2]	0.063
RDW CV (%)	13.8 [13.0–15.0]	13.9 [12.9–15.4]	14.2 [13.5–15.9]	13.9 [12.9–14.9]	<0.001
WBC (*1000/microL), median [IQR]	8.0 [5.7–11.2]	6.4 [5.5–10.6]	7.4 [4.9–9.2]	9.2 [6.1–11.8]	<0.001
NLR, median [IQR]	5.85 [3.28–10.56]	4.59 [2.60–8.70]	4.09 [1.93–7.38]	7.88 [5.18–11.89]	<0.001
MLR, median [IQR]	0.54 [0.37–0.82]	0.54 [0.34–0.69]	0.42 [0.32–0.73]	0.57 [0.36–0.99]	<0.001
SII, median [IQR]	1306.5 [636.9–2740.2]	990.0 [427.0–1530.8]	1193.4 [719.1–2156.9]	1257.9 [805.8–2735.9]	<0.001
hs-CRP (mg/dL), median [IQR]	7.2 [2.3–15.7]	2.0 [0.74–1.2]	6.2 [2.8–19.8]	9.4 [1.9–16.7]	<0.001
Ferritin (ng/mL), median [IQR]	536 [235–1285]	475 [193–955]	536 [235–1356]	771 [325–1711]	<0.001
Presepsin (pg/mL), median [IQR]	356 [193–696]	285 [188–506]	150 [91–384]	494 [252–1066]	<0.001
Creatinine (mg/dL), median [IQR]	0.9 [0.8–1.2]	1.0 [0.7–1.2]	0.9 [0.7–1.2]	0.9 [0.8–1.2]	<0.001
ALAT (U/L), median [IQR]	33 [21–55]	21 [13–33]	30 [18–66]	29 [14–48]	0.072
hs-TnI (ng/L), median [IQR]	10.2 [2.4–44.5]	8.5 [2.0–28.8]	4.1 [1.5–17.7]	18.1 [5.6–81.7]	<0.001
BNP (pg/mL), median [IQR]	204 [55.8–581]	260 [53.2–758]	204 [26–492]	185 [59–592]	0.840
NT-pro BNP (pg/mL), median [IQR]	844.5 [235.8–3207.3]	698.0 [214.3–2256.3]	416.5 [120.5–1380.7]	1726.5 [555.0–4935.8]	<0.001
Hospitalization (d), median [IQR]	12 [4–16]	14 [6–17]	13 [4–16]	7 [3–15]	<0.001

*, comparison between outcome groups; **, percentage of total patients; IQR, interquartile range; CCI, Charlson comorbidity index; NEWS, National Early Warning Score; SBP, systolic blood pressure; WBC, white blood cells; MLR, monocyte-to-lymphocyte ratio; SII, systemic inflammatory index; hs-CRP, high sensitivity C-reactive protein; ALAT, alanine aminotransferase.

**Table 2 diagnostics-12-01373-t002:** Cardiac, selected inflammation biomarkers, and age between outcome groups in relation with the virus variant.

Alpha Variant N = 509 (52%)	Parameter, Median [IQR]	Survivors without Sequelae N = 279 (28.5%)	Post-Acute COVID-19 N = 104 (10.6%)	30-Day Mortality N = 126 (12.9%)	*p* Value *^,a^
	NT-proBNP (pg/mL)	691 [211–2256]	398 [127–1266]	2001 [692–4810]	<0.001
	hs-TnI (ng/L)	8.3 [2.1–31.4]	7.5 [2.3–22.3]	17.4 [6.7–62.2]	0.017
	hs-CRP (mg/dL)	4.4 [1.1–10.4]	5.0 [1.6–9.6]	6.7 [2.3–15.8]	<0.001
	RDW CV (%)	13.8 [12.9–15.0]	13.3 [12.7–14.1]	14.7 [13.5–16.4]	<0.001
	NLR	3.9 [2.3–6.6]	4.0 [2.3–6.5]	6.4 [3.7–10.3]	<0.001
	MLR	0.5 [0.3–0.7]	0.5 [0.3–0.7]	0.7 [0.4–0.9]	<0.001
	SII	919 [454–1725]	862 [459–1631]	1219 [636–2591]	0.010
	Ferritin (ng/mL)	353 [164–789]	467 [223–872]	483 [177–1261]	0.050
	Presepsin (pg/mL)	285 [161–493]	226 [144–421]	560 [311–1359]	<0.001
	Fibrinogen (mg/dL)	413 [326–495]	426 [371–487]	411 [306–505]	0.594
	WBC (*1000/microL)	6.92 [4.99–9.41]	6.59 [4.91–8.71]	10.20 [6.82–14.32]	<0.001
	Age (years)	66 [54–76]	68 [57–73]	73 [65–81]	<0.001
**Beta** **Variant** **N = 120 (12.3%)**	**Parameter,** **Median [IQR]**	**Survivors without Sequelae** **N = 49 (5.0%)**	**Post-Acute COVID-19** **N = 29 (3.0%)**	**30-Day Mortality** **N = 42 (4.3%)**	***p* Value *^,a^**
	NT-pro BNP (pg/mL)	582 [105–1489]	469 [101–2865]	2182 [200–5028]	0.185
	hs-TnI (ng/L)	2.8 [0.1–10.6]	3.3 [0.1–18.1]	16.4 [0.1–79.4]	0.134
	hs-CRP (mg/dL)	5.2 [2.2–16.7]	6.5 [1.6–16.8]	10.8 [2.1–20.7]	0.513
	RDW CV (%)	14.1 [12.8–16.2]	13.5 [12.8–14.5]	14.7 [13.7–16.0]	0.061
	NLR	5.8 [2.9–12.2]	4.9 [3.4–11.4]	9.9 [4.8–14.8]	0.110
	MLR	0.5 [0.3–0.8]	0.5 [0.4–0.9]	0.5 [0.3–0.9]	0.856
	SII	1449 [624–3386]	1193 [567–2372]	1927 [846–3892]	0.337
	Ferritin (ng/mL)	422 [147–853]	468 [131–1629]	856 [283–1608]	0.086
	Presepsin (pg/mL)	307 [202–509]	201 [147–525]	874 [479–1105]	0.016
	Fibrinogen (mg/dL)	444 [366–567]	442 [342–579]	433 [359–567]	0.923
	WBC (*1000/microL)	7.93 [5.84–11.86]	8.16 [5.26–11.14]	8.73 [5.75–13.38]	0.507
	Age (years)	70 [59–81]	68 [59–74]	74 [66–80]	0.107
**Delta** **Variant** **N = 349 (35.7%)**	**Parameter,** **Median [IQR]**	**Survivors without Sequelae** **N = 108 (11.0%)**	**Post-Acute COVID-19** **N = 66 (6.8%)**	**30-Day Mortality** **N = 175 (17.9%)**	***p* Value *^,a^**
	NT-pro BNP (pg/mL)	850 [306–3480]	468 [92–1195]	1394 [555–4957]	0.003
	hs-TnI (ng/L)	13.4 [3.8–38.9]	3.8 [0.1–12.6]	20.7 [5.2–116.0]	0.003
	hs-CRP (mg/dL)	11.3 [4.3–20.9]	10.2 [5.73–19.4]	14.9 [7.3–22.9]	0.028
	RDW CV (%)	13.7 [13.1–14.4]	13.6 [12.8–14.6]	14.1 [13.3–15.1]	0.019
	NLR	6.6 [4.7–13.5]	6.3 [4.3–9.6]	10.0 [5.9–17.2]	<0.001
	MLR	0.6 [0.4–0.9]	0.5 [0.4–0.7]	0.6 [0.5–1.0]	0.010
	SII	1744 [930–3545]	1754 [898–2577]	2263 [1133–4533]	0.009
	Ferritin (ng/mL)	711 [499–1755]	693 [410–1.755]	950 [500–2072]	0.158
	Presepsin (pg/mL)	457 [264–651]	570 [461–1174]	735 [300–1245]	0.228
	Fibrinogen (mg/dL)	543 [443–606]	492 [410–573]	514 [441–584]	0.310
	WBC (*1000/microL)	8.31 [6.04–10.72]	7.99 [5.70–10.53]	9.41 [6.79–13.95]	0.002
	Age (years)	71 [62–79]	64 [53–73]	72 [65–81]	<0.001

N, number of patients; %, of total; IQR, interquartile range; *, comparison between groups; ^a^ *p*-value estimated using the Kruskal–Wallis’ test; NT-pro BNP, N-terminal pro-B-type natriuretic peptide; hs-TnI, high-sensitivity TnI; hs-CRP, high sensitivity C-reactive protein; RDW-CV, red cell distribution width, coefficient of variation; NLR, neutrophil-to-lymphocyte ratio; MLR, monocyte-to-lymphocyte ratio; SII, systemic inflammatory index.

**Table 3 diagnostics-12-01373-t003:** Biomarker/clinical indicator performance to predict 30-day mortality in noncritically ill COVID-19 patients.

Biomarker/Clinical Indicator ^	AUC (95% CI)	Sensitivity at Youden’s Index	Specificity at Youden’s Index
NLR * (n = 555)	0.79 (0.74–0.83)	0.65	0.41
MLR * (n = 556)	0.73 (0.68–0.77)	0.70	0.50
Presepsin (n = 434)	0.73 (0.68–0.78)	0.70	0.36
hs-CRP * (n = 532)	0.72 (0.67–0.77)	0.74	0.65
hs-TnI * (n = 254)	0.70 (0.59–0.81)	0.63	0.48
SII * (n = 555)	0.69 (0.64–0.75)	0.55	0.40
WBC * (n = 568)	0.68 (0.63–0.73)	0.76	0.45
NT-pro BNP (n = 978)	0.66 (0.62–0.70)	0.65	0.40
RDW * (n = 549)	0.64 (0.60–0.70)	0.68	0.51
Fibrinogen * (n = 349)	0.55 (0.48–0.63)	0.44	0.40
Ferritin (n = 798)	0.53 (0.46–0.61)	0.73	0.64
IL-6 (n = 39)	0.53 (0.49–0.58)	0.91	0.85
NEWS 2 (n = 978)	0.44 (0.40–0.48)	0.31	0.44
Age (n = 978)	0.40 (0.36–0.44)	0.56	0.68
CCI (n = 978)	0.37 (0.33–0.41)	0.32	0.55

^, blood work was not collected from all participants for all tests, with the number included in each model listed; AUC, area under the curve; *, values obtained before discharge; hs-CRP, high sensitivity C-reactive protein; MLR, monocyte-to-lymphocyte ratio; NLR, neutrophil-to-lymphocyte ratio; SII, systemic inflammatory index; RDW, red cell distribution width; WBC, white blood cells count; IL-6, interleukin-6; NT-pro BNP, N-terminal B-type natriuretic peptide; hs-TnI, high sensitivity cardiac troponin I; CCI, Charlson comorbidity index; NEWS, National Early Warning Score.

**Table 4 diagnostics-12-01373-t004:** Biomarker/clinical indicator performance to predict post-acute long COVID-19 symptoms in noncritically ill COVID-19 patients.

Biomarker/Clinical Indicator ^	AUC (95% CI)	Sensitivity at Youden’s Index	Specificity at Youden’s Index
Il-6 (n = 39)	0.53 (0.49–0.58)	0.91	0.85
Fibrinogen * (n = 349)	0.52 (0.44–0.59)	0.48	0.53
Ferritin * (n = 372)	0.50 (0.43–0.57)	0.48	0.42
NT-pro BNP * (n = 638)	0.45 (0.36–0.55)	0.70	0.80
WBC * (n = 568)	0.44 (0.38–0.50)	0.42	0.55
NEWS 2 (n = 978)	0.44 (0.40–0.48)	0.31	0.44
hs-CRP * (n = 532)	0.41 (0.36–0.47)	0.59	0.71
SII * (n = 555)	0.40 (0.35–0.46)	0.49	0.64
Age (n = 978)	0.40 (0.36–0.44)	0.56	0.68
RDW * (n = 549)	0.39 (0.33–0.45)	0.55	0.74
NLR * (n = 555)	0.38 (0.32–0.43)	0.84	0.71
hs-TnI * (n = 254)	0.38 (0.30–0.47)	0.68	0.79
MLR * (n = 556)	0.37 (0.32–0.43)	0.49	0.68
CCI (n = 978)	0.37 (0.33–0.41)	0.32	0.55
Presepsin (n = 434)	0.36 (0.30–0.42)	0.65	0.44

^, blood work was not collected from all participants for all tests, with the number included in each model listed; AUC, area under the curve; *, values obtained before discharge; hs-CRP, high sensitivity C-reactive protein; MLR, monocyte-to-lymphocyte ratio; NLR, neutrophil-to-lymphocyte ratio; SII, systemic inflammatory index; RDW, red cell distribution width; WBC, white blood cells count; IL-6, interleukin-6; NT-pro BNP, N-terminal B-type natriuretic peptide; hs-TnI, high sensitivity cardiac troponin I; CCI, Charlson comorbidity index; NEWS, National Early Warning Score.

**Table 5 diagnostics-12-01373-t005:** Cox proportional-hazards model assessing the relationship between NT-pro BNP and mortality during follow-up adjusted for relevant covariates.

Variable	Univariable		Multivariable	
	HR (95%CI)	*p* Value	HR (95%CI)	*p* Value
NT-pro BNP (per log. unit)	1.96 (1.54–2.50)	<0.001	1.63 (1.10–2.42)	0.01
Age (per 10 years)	1.03 (1.02–1.04)	<0.001	1.03 (1.00–1.05)	0.04
Male sex	0.82 (0.61–1.08)	0.15	1.03 (0.62–1.69)	0.92
NEWS2	1.16 (1.11–1.21)	<0.001	0.95 (0.85–1.06)	0.35
Admission SaO_2_ < 90%	0.46 (0.35–0.61)	<0.001	0.50 (0.26–0.97)	0.04
History of AD	1.03 (0.71–1.49)	0.89	0.72(0.40–1.32)	0.29
History of HT	0.87 (0.61–1.25)	0.44	0.79 (0.48–1.28)	0.33
History of DM	1.78 (1.27–2.49)	0.001	1.37 (0.32–5.81)	0.66
History of CKD	0.39(0.24–0.64)	<0.001	0.51 (0.23–1.14)	0.10
History of HF	0.72 (0.49–1.04)	0.08	2.03 (0.98–4.20)	0.05
SBP (per 10 mmHg)	0.99 (0.98–1.01)	0.08	0.16 (0.01–3.45)	0.24
AHF upon admission	0.81 (0.57–1.15)	0.24	1.29 (0.58–2.90)	0.53
Mechanical ventilation	0.84 (0.50–1.41)	0.50	2.06 (0.90–4.70)	0.08
Corticotherapy	0.63 (0.45–0.87)	0.006	0.73 (0.45–1.17)	0.19
Virus variant (Delta)	2.01 (1.66–2.43)	<0.001	1.68 (1.27–2.24)	<0.001
CCI	1.16 (1.08–1.23)	<0.001	1.06 (0.92–1.21)	0.42

HR, hazard ratio; CI, confidence interval; SE, standard error; NEWS2, National Early Warning Score; AD, atherosclerotic disease; HT, arterial hypertension; DM, diabetes mellitus; CKD, chronic kidney disease; HF, heart failure; SBP, systolic blood pressure; AHF, acute heart failure; CCI, Charlson comorbidity index.

## Data Availability

The data presented in this study are available in this article and Appendix A.

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
