# Peer review of "Inflammatory and Cardiac Biomarkers in Relation with Post-Acute COVID-19 and Mortality: What We Know after Successive Pandemic Waves"

_diagnostics, 2022, doi:10.3390/diagnostics12061373_

Round 1

Reviewer 1 Report

1) Abstract. L 32-42. Results. Post-acute COVID-19 was diagnosed in 20.3% and all-cause 30-days mortality was 35.1% among 978  patients infected with variants of concern. Significant correlations with the outcomes were recorded for inflammatory and cardiac biomarkers, after adjustments for significant confounders. Neutrophil-to-lymphocyte ratio, monocyte-to-lymphocyte ratio, C-reactive protein and cardiac biomarkers  showed predictive value for the outcomes. Cox proportional‐hazards model confirmed that NT‐  proBNP was independently associated with mortality. NT‐proBNP remained independently asso-  ciated with 30-days mortality during follow‐up (1.29 [95%CI, 1.07-1.56], p = 0.007) after adjustment for confounders. Conclusion. Inflammation and cardiac biomarkers, determined upon admission  and predischarge, in a cohort of hospitalized non-critically ill COVID-19 patients throughout successive pandemic waves, showed a predictive value for post-acute COVID-19 and 30-days mortality

Please add the most important statistical values to support the results and conclusions.

2) Introduction. L52-56. The persistence of symptoms or development of sequelae beyond 3 or 4 weeks from the onset  of acute symptoms of COVID-19 defines post-acute COVID-19 [3,4]. Given the variable severity of COVID-19, and its unpredictable clinical course, prognostic biomarkers would  be invaluable when triaging patients to identify high-risk patients upon hospital admission or discharge. Please improve this paragraph and add these references:

a- Coronavirus disease 2019 (COVID-19): An overview of the immunopathology, serological diagnosis and management. Scand J Immunol. 2021;93(4):e12998. doi:10.1111/sji.12998

b- Interstitial Lung Disease at High Resolution CT after SARS-CoV-2-Related Acute Respiratory Distress Syndrome According to Pulmonary Segmental Anatomy. J. Clin. Med. 202110, 3985. https://doi.org/10.3390/jcm10173985

3) Introduction. L76-78.  We undertook this study to directly compare the independent prognostic value of  inflammatory and cardiac biomarkers in non-critically ill COVID-19 patients infected with  different variants of SARS-CoV-2 in relation with post-acute COVID-19 and mortality. Please improve the description of study aim.

4) Could you please specify if different treatment regimes of patients change the results?

5) 4. Discussion L303-307 COVID-19 remains a clinical challenge for every practitioner. World Health Organi- zation have mentioned, among SARS-CoV-2 variants of concern, Alpha (B.1.1.7), Beta (B.1.351), Gamma (P.1), and Delta (B.1.617.2). Alpha, Beta, Gamma, and Delta variants are 306 all more serious than the wild-type virus in terms of hospitalization and mortality, with the Beta and Delta variants having a higher risk than the Alpha and Gamma variants [17]. Could you please underline here the most important results of study?

6) 5. Conclusions L411-421. Inflammation and cardiac biomarkers, when performed on admission and predis-charge, in a cohort of hospitalized patients for non-critically ill COVID-19 throughout successive pandemic waves showed a moderate predictive value for 30-days mortality and  modest predictive value for post-acute long COVID-19. Age and routine clinical scores  (i.e., CCI and NEWS2 score) showed also a poor predictive role. Among inflammation  biomarkers, NLR, hs-CRP and MLR had the best performance. Also, NT-pro BNP showed  a good predictive role for 30-days mortality, irrespective of variant of concern involved. Assessing these biomarkers may represent a surrogate of invasive monitoring in a context  of a poor resource setting, support the tailoring of medical therapy and to guide allocation  of available resources. Further large prospective studies should validate the additional value of these biomarkers to routinely collected clinical information.  Please underline the clinical implication of the study.

Author Response

Dear Editor

We thank the reviewer for the observations and suggestions. As a result, we’ve made changes in the manuscript in the sections suggested. We used track changes and we highlighted the text in red color. We hope that these changes will increase the value of our manuscript and it will be suitable for publication.

Kind regards, Assoc. Prof. Catalina Lionte (correspondent author)

Comments and Suggestions for Authors

First, we checked English language and style, as suggested.

1) Abstract. L 32-42. Results. Post-acute COVID-19 was diagnosed in 20.3% and all-cause 30-days mortality was 35.1% among 978  patients infected with variants of concern. Significant correlations with the outcomes were recorded for inflammatory and cardiac biomarkers, after adjustments for significant confounders. Neutrophil-to-lymphocyte ratio, monocyte-to-lymphocyte ratio, C-reactive protein and cardiac biomarkers  showed predictive value for the outcomes. Cox proportional‐hazards model confirmed that NT‐  proBNP was independently associated with mortality. NT‐proBNP remained independently asso-  ciated with 30-days mortality during follow‐up (1.29 [95%CI, 1.07-1.56], p = 0.007) after adjustment for confounders. Conclusion. Inflammation and cardiac biomarkers, determined upon admission  and predischarge, in a cohort of hospitalized non-critically ill COVID-19 patients throughout successive pandemic waves, showed a predictive value for post-acute COVID-19 and 30-days mortality

Issue: Please add the most important statistical values to support the results and conclusions.

Answer: We’ve included in the abstract the informations required.

2) Introduction. L52-56. The persistence of symptoms or development of sequelae beyond 3 or 4 weeks from the onset  of acute symptoms of COVID-19 defines post-acute COVID-19 [3,4]. Given the variable severity of COVID-19, and its unpredictable clinical course, prognostic biomarkers would  be invaluable when triaging patients to identify high-risk patients upon hospital admission or discharge.

Issue: Please improve this paragraph and add these references:

a- Coronavirus disease 2019 (COVID-19): An overview of the immunopathology, serological diagnosis and management. Scand J Immunol. 2021;93(4):e12998. doi:10.1111/sji.12998

b- Interstitial Lung Disease at High Resolution CT after SARS-CoV-2-Related Acute Respiratory Distress Syndrome According to Pulmonary Segmental Anatomy. J. Clin. Med. 202110, 3985. https://doi.org/10.3390/jcm10173985

Answer: We’ve improved the introduction paragraphs and we addedd the references suggested.

3) Introduction. L76-78.  We undertook this study to directly compare the independent prognostic value of  inflammatory and cardiac biomarkers in non-critically ill COVID-19 patients infected with  different variants of SARS-CoV-2 in relation with post-acute COVID-19 and mortality.

Issue: Please improve the description of study aim.

Answer: We included a better description of the study aim in the introduction section.

4) Issue: Could you please specify if different treatment regimes of patients change the results?

Answer: We’ve included in the results section the influence of the therapy regimens on the outcomes (page six, last paragraph): “In terms of therapy, administration of corticotherapy was significantly correlated with protection from both post-acute COVID-19 (63%, vs 37%, p < 0.001) and 30-days mortality (80.8%, vs 19.2%, p < 0.001), while antiviral therapy administration was correlated with absence of long COVID (79.7%, vs 20.3%, p < 0.001) and mortality (80.7%, vs 19.3%, p 0.006)”.

5) 4. Discussion L303-307 COVID-19 remains a clinical challenge for every practitioner. World Health Organi- zation have mentioned, among SARS-CoV-2 variants of concern, Alpha (B.1.1.7), Beta (B.1.351), Gamma (P.1), and Delta (B.1.617.2). Alpha, Beta, Gamma, and Delta variants are 306 all more serious than the wild-type virus in terms of hospitalization and mortality, with the Beta and Delta variants having a higher risk than the Alpha and Gamma variants [17].

Issue: Could you please underline here the most important results of study?

Answer: We included a paragraph which underlines the most important results of the sudy, as suggested.

6) 5. Conclusions L411-421. Inflammation and cardiac biomarkers, when performed on admission and predis-charge, in a cohort of hospitalized patients for non-critically ill COVID-19 throughout successive pandemic waves showed a moderate predictive value for 30-days mortality and  modest predictive value for post-acute long COVID-19. Age and routine clinical scores  (i.e., CCI and NEWS2 score) showed also a poor predictive role. Among inflammation  biomarkers, NLR, hs-CRP and MLR had the best performance. Also, NT-pro BNP showed  a good predictive role for 30-days mortality, irrespective of variant of concern involved. Assessing these biomarkers may represent a surrogate of invasive monitoring in a context  of a poor resource setting, support the tailoring of medical therapy and to guide allocation  of available resources. Further large prospective studies should validate the additional value of these biomarkers to routinely collected clinical information. 

Issue: Please underline the clinical implication of the study.

Answer: We underlined in the conclusion section the clinical implication of the study, as suggested.

Reviewer 2 Report

Lionte et. al reported work, "Inflammatory and Cardiac Biomarkers in Relation with Post-Acute COVID-19 and Mortality – What Do We Know after Successive Pandemic Waves" interesting however manuscript needs revision for the publication in Diagnostics. 

In general Covid recovered patients can be again infected with another strain, hence multiple strain affected patients post-covid symptoms need to be specified.  

1) Some of the issues such as major post-covid symptoms, in relation with all strains need to be briefly addressed. Because author says " non-critically ill". 

2) What about the post-covid symptoms If the patient affected by multiple strains. Briefly it need to be addressed. 

3) Recent references related to post-covid symptoms and clinical diagnosis results need to be conjoined with current work specially in the results and discussion.

4) Most of the graphs are not clearly visible, I rather recommend to author use Origin software to draw the figures. 

5) Authors need to briefly explains the roles of biomarkers such as NLR, hs-CRP and MLR in post-covid patients along with symptoms (non-critically ill). If asymptotic then how these biomarkers differ from multiple strain attacked patients.  

Author Response

Dear Editor

We thank the reviewer for the observations and suggestions. As a result, we’ve made changes in the manuscript in the sections suggested. We used track changes and we highlighted the text in red color. We hope that these changes will increase the value of our manuscript and it will be suitable for publication.

Kind regards, Assoc. Prof. Catalina Lionte (correspondent author)

Comments and Suggestions for Authors

First, we checked English language and style, as suggested.

Lionte et. al reported work, "Inflammatory and Cardiac Biomarkers in Relation with Post-Acute COVID-19 and Mortality – What Do We Know after Successive Pandemic Waves" interesting however manuscript needs revision for the publication in Diagnostics. 

Issue: In general Covid recovered patients can be again infected with another strain, hence multiple strain affected patients post-covid symptoms need to be specified.  

Answer: We specified clearly that in patients with post-acute COVID-19 we did not prove a reinfection with another viral strain. Also, one of the exclusion criteria was readmission during the clinical study period and these patients were not included in this analysis. However, in the patients readmitted in our department we did not prove a reinfection with another strain.

Issue: 1) Some of the issues such as major post-covid symptoms, in relation with all strains need to be briefly addressed. Because author says " non-critically ill". 

Answer: we described the correlation between the viral strain and the occurrence of post-acute COVID-19 and the type of symptoms recorded in our cohort, and we added this information in the result section.

Issue: 2) What about the post-covid symptoms If the patient affected by multiple strains. Briefly it need to be addressed. 

Answer: We did not record, in our cohort, in the time frame analyzed, patients infected by multiple strains. We’ve included an additional comment about this issue in the results section.

3) Recent references related to post-covid symptoms and clinical diagnosis results need to be conjoined with current work specially in the results and discussion.

Answer: We included in the discussion section recent references related to post-Covid symptoms and clinical diagnosis, and we discussed our results compared with these references, as suggested.

4) Most of the graphs are not clearly visible, I rather recommend to author use Origin software to draw the figures. 

Answer: We had to format the figures according to the journal requirements. As a result, although the resolution is 300 dpi, the small dimension of the figure makes it unclear. Unfortunately, we did not have the software recommended. It will be possible, in the final draft of the manuscript, to enlarge the figures, to provide more clear visibility.

5) Authors need to briefly explains the roles of biomarkers such as NLR, hs-CRP and MLR in post-Covid patients along with symptoms (non-critically ill). If asymptotic then how these biomarkers differ from multiple strain attacked patients.  

Answer: we included the correlation of the above-mentioned biomarkers with the occurrence of post-acute COVID-10 symptoms in the results section, and in the supplementary table S1, which we included in the supplementary files. We did not record patients infected with multiple strains in our cohort.

Reviewer 3 Report

Review of “Inflammatory and cardiac biomarkers in relation with post-acute COVID-19 and mortality – what do we know after successive pandemic waves” (diagnostics-1726111)

This study investigated the impact of Inflammatory and cardiac biomarkers on mortality and post-acute COVID-19 of COVID-19 in patients with COVID-19. The concept of this study is interesting, however, several problems to be solved.

  1. The study design was unclear from Title and/or Abstract. Did the authors really describe the manuscript according to STROBE guideline?
  2. To show inclusion and exclusion flows is needed. How many were included and how many were excluded are unclear.
  3. This reviewer is confused regarding the definitions of inclusion and exclusion. “Patients who died during hospitalization or transferred to other departments, or intensive care unit, patients who were alive post-discharge, but not completed the 30-days follow up before the lock of the database, patients discharged against medical advice, and patients with incomplete data were excluded.” Were the patients who died within 30 days excluded?
  4. Furthermore, this reviewer thinks that this exclusion criterion may bias the study participants considerably, because transferred to other departments, or intensive care unit were excluded.
  5. Definition of post-acute COVID-19 and non-critically ill were unclear.
  6. This reviewer gets the impression that it is not correct to analyze post-acute COVID-19 and mortality patients together in Table 4.
  7. Supplemental data cannot be downloaded.

Author Response

            Dear Editor

We thank the reviewer for the observations and suggestions. As a result, we’ve made changes in the manuscript in the sections suggested. We used track changes and we highlighted the text in red/blue color. We hope that these changes will increase the value of our manuscript and it will be suitable for publication.

Kind regards, Assoc. Prof. Catalina Lionte (correspondent author)

Comments and Suggestions for Authors

First, we checked English language and style, as suggested.

Review of “Inflammatory and cardiac biomarkers in relation with post-acute COVID-19 and mortality – what do we know after successive pandemic waves” (diagnostics-1726111)

This study investigated the impact of Inflammatory and cardiac biomarkers on mortality and post-acute COVID-19 of COVID-19 in patients with COVID-19. The concept of this study is interesting, however, several problems to be solved.

  1. The study design was unclear from Title and/or Abstract. Did the authors really describe the manuscript according to STROBE guideline?

Answer: We completed the study design in the abstract. We wish to maintain, if it’s possible, the title proposed, without mentioning in the title the study type.

  1. To show inclusion and exclusion flows is needed. How many were included and how many were excluded are unclear.

Answer: The study flowchart is presented in Supplementary figure S1.

  1. This reviewer is confused regarding the definitions of inclusion and exclusion. “Patients who died during hospitalization or transferred to other departments, or intensive care unit, patients who were alive post-discharge, but not completed the 30-days follow up before the lock of the database, patients discharged against medical advice, and patients with incomplete data were excluded.” Were the patients who died within 30 days excluded?

Answer: We included patients who died within 30-days from discharge in the analysis (Table 1). The patients who were not included were those for which we had no information regarding their status (i.e., alive, without symptoms; alive with post-COVID symptoms, or dead within 30 days post discharge).

  1. Furthermore, this reviewer thinks that this exclusion criterion may bias the study participants considerably, because transferred to other departments, or intensive care unit were excluded.

Answer: We aimed to characterize a cohort of non-critically ill COVID-19 patients. Those who were transferred to other departments or ICU were patients who developed shortly a severe/critically form of COVID-19. As a result of this observation, we described clearer the aim of the study and the type of population included in the analysis.

  1. Definition of post-acute COVID-19 and non-critically ill were unclear.

Answer: The definition for post-acute COVID-19 is based on references 3 (Nat Med. 2021;27, 601-615) and 4 (https://www.cdc.gov/coronavirus/2019-ncov/long-term-effects/index.html). We included in the introduction a definition of non-critically ill in the study setting and participants section, and we clearly defined a critically COVID-19 in the introduction section.

  1. This reviewer gets the impression that it is not correct to analyze post-acute COVID-19 and mortality patients together in Table 4.

Answer: In table 4 we analyzed biomarkers’ ability to predict post-acute long COVID symptoms (Table 4. Biomarker/clinical indicator performance to predict post-acute long COVID symptoms in noncritically ill COVID-19 patients)

  1. Supplemental data cannot be downloaded.

Answer: it is unfortunate that you could not download supplementary files. This issue might be resolved by the Editors.

Round 2

Reviewer 2 Report

Revised manuscript can be acceptable in current form. 

Author Response

Dear Reviewer

We would like to thank you for your observations and suggestions, which improved the quality of our manuscript. We would review again the manuscript for English minor spell check errors. Kind regards, Assoc. Prof. Catalina Lionte

Reviewer 3 Report

Several problems have not been answered well.

1. Unable to open supplemental files. Thus, this reviewer cannot evaluate this article.  

2. Definition of post-acute COVID-19 and non-critically ill should be shown in the Methods section.

3. This reviewer feels that the purpose of this study are not consistent, since it is only a contradiction to include the patients who died within 30 days but not those who were critically ill.

4. Table 4 is supposed to be for the non-critically ill, but this reviewer did not understand why data for all patients is used. (Ex. CCI, NEWS and age are described as n = 978)

Author Response

Dear Editor

We thank the reviewer for the observations and suggestions. As a result, we’ve made changes in the manuscript in the sections suggested. We used track changes and we highlighted the text in red color. We hope that with this changes and answers provided, we have made clear the issues raised by the reviewer. We hope that in this form, our manuscript will be suitable for publication.

Kind regards, Assoc. Prof. Catalina Lionte (correspondent author)

Comments and Suggestions for Authors

First, we checked English language and style, as suggested.

Several problems have not been answered well.

  1. Unable to open supplemental files. Thus, this reviewer cannot evaluate this article.  

Answer: We contacted the Editor of the Journal, we told him this problem and we were assured that you will be contacted directly by the Editor with this problem and the supplementary files will be available to you.

  1. Definition of post-acute COVID-19 and non-critically ill should be shown in the Methods section.

Answer: We defined in the Methods section the post-acute COVID-19 and non-critically ill COVID-19 in the Methods section, as suggested, for better clarification of the population included.

  1. This reviewer feels that the purpose of this study are not consistent, since it is only a contradiction to include the patients who died within 30 days but not those who were critically ill.

Answer: Many studies assessed biomarkers in critically ill COVID-19 patients. However, scarce information exists about hospitalized patients with non-critical forms of COVID-19. This was our aim, to characterize this population, COVID-19 patients who are hospitalized with mild, moderate or severe illness, with associated comorbidities, in terms of biomarkers and their prognostic role. The patients who developed during hospitalization a critical form of COVID-19 and required ICU care, or were transferred to Pneumology departments, for example, were not included, because we did not wish to replicate the same research regarding critically ill COVID-19 patients. Also, data regarding their evolution and outcomes were not fully available to be analyzed.

  1. Table 4 is supposed to be for the non-critically ill, but this reviewer did not understand why data for all patients is used. (Ex. CCI, NEWS and age are described as n = 978)

Answer: All patients included in this cohort were non-critically ill COVID-19 cases, they were hospitalized in a medical ward. We used for this analysis data which were available for all patients included (n=978), such as age, CCI (Charlson comorbidity index, etc), and biomarkers’ values upon discharge from the hospital, that is why we signaled with an * and we mentioned distinctly in brackets the number of the patients with this data available (as we stated in the Methods section, the determination of blood tests during hospitalization or pre-discharge was done upon attending physician request).

Round 3

Reviewer 3 Report

This reviewer ubderstood the content well. No further comments.

Author Response

Dear Reviewer

First, we would like to thank you for your observations and suggestions which improved our manuscript.

Comments and Suggestions for Authors

This reviewer ubderstood the content well. No further comments.

We thank you for the acceptance of the revised manuscript. Kind regards, Assoc. Prof. Catalina Lionte (correspondent author)